# Phosphorus-Solubilizing Capacity of *Mortierella* Species Isolated from Rhizosphere Soil of a Poplar Plantation

**DOI:** 10.3390/microorganisms10122361

**Published:** 2022-11-29

**Authors:** Yue Sang, Long Jin, Rui Zhu, Xing-Ye Yu, Shuang Hu, Bao-Teng Wang, Hong-Hua Ruan, Feng-Jie Jin, Hyung-Gwan Lee

**Affiliations:** 1College of Biology and the Environment, Co-Innovation Center for Sustainable Forestry in Southern China, Nanjing Forestry University, 159 Longpan Road, Nanjing 210037, China; 2Cell Factory Research Centre, Korea Research Institute of Bioscience & Biotechnology (KRIBB), Daejeon 34141, Republic of Korea

**Keywords:** *Mortierella*, phosphorus-solubilizing fungi, phosphate-solubilizing microorganisms, phosphorus-solubilizing capacity

## Abstract

Phosphorus is one of the main nutrients necessary for plant growth and development. Phosphorus-dissolving microorganisms may convert insoluble phosphorus in soil into available phosphorus that plants can easily absorb and utilize. In this study, four phosphorus-solubilizing fungi (L3, L4, L5, and L12) were isolated from the rhizosphere soil of a poplar plantation in Dongtai, Jiangsu Province, China. Phylogenetic analysis based on the internal transcribed spacer (ITS) and large subunit (LSU) of the ribosomal DNA sequences showed that the ITS and 28S sequences of isolates were the most similar to those of *Mortierella*. Morphological observation showed that most colonies grew in concentric circles and produced spores under different culture conditions. These results and further microscopic observations showed that these isolated fungi belonged to the genus *Mortierella*. Pikovskaya (PKO) medium, in which tricalcium phosphate was the sole phosphorus source, was used to screen strain L4 with the best phosphorus-solubilizing effect for further study. When the carbon source was glucose, the nitrogen source was ammonium chloride, the pH was 5, and the available phosphorus content was the highest. By exploring the possible mechanism of phosphorus release by phosphorus-solubilizing fungi, it was found that strain L4 produces several organic acids, such as oxalic acid, lactic acid, acetic acid, succinic acid, tartaric acid, malic acid, and citric acid. At 24 h, the alkaline phosphatase and acid phosphatase activities reached 154.72 mol/(L·h) and 120.99 mol/(L·h), respectively.

## 1. Introduction

Phosphorus is an essential nutrient for plant growth and development [1]. Phosphorus plays important roles in photosynthesis, respiration, energy production and storage, nucleic acid biosynthesis, and other physiological and biochemical processes in plants [2,3]. However, bioavailable phosphorus concentrations in soils are generally low, with typically only 0.1–0.5% of the total phosphorus in a soluble form available to plants [4]. Extensive use of fertilizers has many adverse effects on the environment and soil microbial populations [5], causing soil pollution and water eutrophication [6,7]. Microorganisms play a critical role in soil phosphorus cycle, regulating phosphorus availability and promoting phosphorus uptake by plants [8,9]. Phosphorus-solubilizing microorganisms can dissolve potential insoluble phosphorus in soil and convert it into soluble phosphorus for plant absorption and utilization. 

In forest and agricultural management practices, phosphate-solubilizing microorganisms (PSMs) have been considered as a new alternative to phosphate fertilizers. In studies of a variety of PSMs, including *Aspergillus*, *Penicillium*, and *Pseudomonas* [10,11,12,13], it has been found that insoluble phosphate may be transformed to a soluble form via acidification, chelation, and exchange reactions. The production of organic acids such as citric, oxalic, succinic, and lactic, which may convert insoluble phosphates into bioavailable phosphates by protonation, is one of the main phosphorus-solubilizing mechanisms of these fungi [14,15]. The decrease in pH in the surrounding environment also increased the dissolution of phosphate. In addition to organic acids, certain PSMs can also produce phosphatases, which are significantly involved in phosphorus metabolism [16]. Currently, PSMs are widely employed to improve agricultural yields in a range of crops, such as wheat [15], soybeans [17], peanuts, and mung beans [18]. 

*Mortierella* is a filamentous fungus commonly found in soil, rhizosphere, and plant tissues, and nearly 125 species have been identified [19]. In 1863, Coemans described the first species of the genus, originally found in mushrooms, and named it *Mortierella polycephala* [20]. The main members of the genus *Mortierella* include *Mortierella alpina*, *Mortierella antarctica*, *Mortierella bisporalis*, *Mortierella indohii*, *Mortierella polycephala*, and *Mortierella reticulata* [19]. Studies have shown that as phosphorus-solubilizing fungi, some *Mortierella* species can transform insoluble phosphorus into soluble phosphorus, thus promoting plant growth and development [21]. In addition, some *Mortierella* spp. are potential producers of arachidonic acid [22], some *Mortierella* spp. may act as nematophagous fungi [23], and some *Mortierella* spp. may produce polyunsaturated fatty acids and exert inductive effects in horticultural plants and fruit trees, increasing their resistance to a variety of pathogens [24]. *Mortierella* fungi present in acidic environments can be used in biotechnological processes such as coal desulfurization and metal bioaccumulation [25], and bioremediation of endosulfan-contaminated sites [26]. 

The aim of this study was to screen high-efficiency PSMs from rhizosphere soil samples of a poplar plantation in Dongtai, Jiangsu Province, China. In this process, four new fungal strains belonging to the genus *Mortierella* were isolated and characterized, and their phosphorus-solubilizing capacity and related mechanisms were further explored. 

## 2. Materials and Methods 

### 2.1. Soil Sampling and Isolation of Phosphate-Solubilizing Fungi

The poplar plantation of Dongtai Forest Farm (120°49′ E, 32°52′ N) is located in Jiangsu Province, China. The rhizosphere soil samples were collected from a depth of 10~20 cm, sealed in plastic bags, and taken back to the laboratory, where they were stored at 4 °C until use. Pikovskaya (PKO) agar medium was prepared in 1 L of water with the following ingredients: glucose, 10 g; (NH_4_)_2_SO_4_, 0.5 g; NaCl, 3 g; MgSO4·7H_2_O, 0.3 g; FeSO_4_·7H_2_O, 0.03 g; MnSO_4_·4H_2_O, 0.03 g; KCl, 0.3 g; Ca_3_(PO_4_)_2_, 5 g; yeast extract, 0.4 g; agar, 18 g; pH 7.0~7.5, and sterilized for 15 min at 121 °C.

Five grams of fresh soil samples were weighed and placed in a conical flask filled with 45 mL sterile water for 1 h shock culture. After a dilution series, soil diluents with different gradients of 10^−5^, 10^−4^, 10^−3^, 10^−2^, and 10^−1^ were obtained. Then, 300 μL of diluents were evenly spread on PKO plates in turn, and incubated at 30 °C for 5–7 days. Streptomycin with a final concentration of 50 μg/mL was added into the PKO medium to inhibit bacterial growth. Colonies of L3, L4, L5, and L12 that appeared on PKO plates after 6 days were subcultured to obtain pure cultures. The isolates L3, L4, L5, and L12 observed from 10^−2^ PKO plates were routinely cultured on PDA plates at 30 °C and stored frozen at −80 °C in 15% (*v/v*) glycerol stock solution. 

### 2.2. Morphological Characterization 

The isolated fungi were inoculated on potato dextrose agar (PDA; BD Difco, Sparks, MD, USA), malt extract agar (MEA; Solarbio, Beijing, China), and synthetic low nutrient agar (SNA; ELITE-MEDIA, China) media and cultured at 30 °C for 7 days to observe their morphological characteristics. An inverted Olympus microscope (IX73, Olympus, Tokyo, Japan) and an FEI Quanta 200 scanning electron microscope (FEI Company, Hillsboro, OR, USA) were used for further microscopic morphological observations. The samples were critically dried with liquid CO_2_ as per standard procedure, and then mounted on aluminum sample stubs with conductive carbon, coated with gold, and photographed using FEI Quanta 200 scanning electron microscopy.

The isolated fungi were cultivated at different temperatures (4, 15, 25, 30, 37 °C) for 7 days, and the growth of the strains was observed every day. The temperature range of strain growth was investigated using PDA medium (BD Difco, Franklin Lakes, NJ, USA).

### 2.3. Sequencing and Phylogenetic Analysis

Genomic DNA was directly extracted from the mycelium of the fungal isolates using a genomic DNA preparation kit (Toyobo, Osaka, Japan), and the ITS region and large subunit (LSU) ribosomal RNA gene were amplified with primer pairs ITS4 and ITS5 [27], and NL1 (5′-GCATATCAATAAGCGGAGGAAAAG-3′) and NL4 (5′-GGTCCGTGTTTCAAGACGG-3′), respectively. The PCR-amplified sequences were compared with ITS and LSU rDNA sequences in the Genbank database of the National Center for Biotechnology Information (NCBI) using the BLAST algorithm [28]. The nucleotide sequences of the fungal isolates were also deposited in the GenBank database, with accession numbers ON038715, ON038743, ON038747, and ON038748 for ITS, and ON045511, ON045513, ON045526, and ON045523 for 28S, respectively. Multiple sequence alignment was accomplished using Clustal_X v.2.1 [29], and Mega 6 software [30] was used to construct a neighbor-joining-based phylogenetic tree to visualize the sequence grouping of new fungal isolates.

### 2.4. Determination of Soluble Phosphorus under Different Culture Conditions

Fungal spore suspension (10^8^ spores /mL) was prepared and 1 mL of spore suspension was added to every 100 mL of PKO liquid medium. Ca_3_(PO_4_)_2_, AlPO_4,_ or FePO_4_ were employed as the only phosphorus sources in the medium to determine the phosphorus solubility of isolated fungi. The strains were cultured in a 150 rpm shaker at 30 °C for 7 days, and the phosphorus-dissolving effect of strains was determined by the molybdenum–antimony resistance colorimetric method. The experiment was repeated three times. The culture supernatant was collected every 24 h to evaluate the dissolved phosphorus concentration and pH value [31]. 

Similarly, fungal spore suspension was cultivated in PKO liquid medium with different carbon sources (glucose, sucrose, maltose, mannitol, or starch), nitrogen sources (ammonium sulfate, potassium nitrate, sodium nitrate, ammonium chloride, or peptone), and initial pH (5, 6, 7, 8, or 9). In this study, glucose and ammonium sulfate in PKO medium were replaced by other carbon and nitrogen sources in equal amounts, respectively. Each treatment was repeated three times. After 7 days, the soluble phosphorus content and the final pH in the culture supernatant were determined. 

### 2.5. Determination of Organic Acids Using HPLC 

PKO medium with Ca_3_(PO_4_)_2_ as the sole phosphorus source was used for culture, and UltiMate3000 high performance liquid chromatography (HPLC) (Thermo Fisher Scientific, Dreieich, Hessen, Germany) was used to detect the composition and content of organic acids in the culture supernatant. Centrifuged at 12,000× *g* at 4 °C for 10 min, the supernatant was obtained and repeatedly filtered using a 0.22 µm nylon filter. The mobile phase was 0.02 mol/L KH_2_PO_4_ buffer (98%) mixed with methanol solution (1%), and the pH was adjusted to 2.6 with 1 mol/L phosphoric acid. The flow rate of mobile phase was 0.5 mL/min, and the total running time was 15 min. The separation of organic acids was carried out on a COSMOSIL 5C18-PAQ column (4.6 × 250 mm; COSMOSIL, Kyoto, Japan) at a temperature of 30 °C with a sample volume of 10 µL. The retention time of each signal was recorded at 210 nm. Standard solutions of the following organic acids were prepared: citric acid, malic acid, succinic acid, oxalic acid, acetic acid, lactic acid, fumaric acid, and tartaric acid (100 mg/L). Three replicates were set for each treatment. The detected organic acids were quantified by comparing their respective peak areas with standard acids [32,33]. The culture supernatant was obtained at 24, 72, 120, and 168 h for detection of organic acids.

### 2.6. Phosphatase Activity Assays

Phosphatase activity was measured using p-nitrophenyl phosphate disodium (PNPP) as a matrix, as proposed by Tabatabal and Bremner [34]. To simplify, 1 mL of 25 mM PNPP and 4 mL of modified universal buffer (MUB) (pH = 6.5; pH = 11) were mixed into 1 mL of culture supernatant and incubated at 37 °C for 1 h. The reaction was then terminated by adding 1 mL CaCl_2_ (0.5 mol/L) and 4 mL NaOH (0.5 mol/L) in succession, and the activity of acid phosphatase (ACP) and alkaline phosphatase (AKP) of each sample was determined using a spectrophotometer at 420 nm after 0, 24, 48, 72, 96, and 120 h culture.

### 2.7. Statistical Analysis

All data were subjected to a one-way analysis of variance (ANOVA) with the software SPSS 25.0. Statistical significance was determined by Duncan’s multiple range test, and *p* < 0.05 was considered to be statistically significant. Data are displayed as mean ± standard error.

## 3. Results 

### 3.1. Phylogenetic Analyses

Four isolates L3, L4, L5, and L12 were isolated from the rhizosphere soil of a poplar plantation, in Jiangsu, China. Combined ITS and 28S gene sequences were compared to determine the phylogenetic relationship between four isolates and close members of *Mortierella* species. Isolate L3 was closely related to *M*. *alpina* CBS 384.71C and formed a monophyletic group with a bootstrap value of 100%, and the other three isolates L4, L5, and L12 formed distinct group with *Mortierella alpina* CBS 219.35 (Figure 1). The strains used for the molecular phylogenetic study are listed in Table 1. Further phylogenetic analysis based on combination genes (ITS-28S rDNA) revealed that the four isolates were most closely related to *Mortierella alpina* (Figure 1). 

### 3.2. Morphological Characterization

For morphological characteristics, fungal strains L3, L4, L5, and L12 were cultured on PDA, MEA, and SNA plates at 30 °C. After 3 days of culture, the micromorphology was further observed by microscope and scanning electron microscope.

L3: The colony growth rate on PDA was moderate, and after 7 days of culture at 30 °C the colony diameter reached 55–60 mm with white and dense mycelia. Both the front and back of the colony were white. The colony reached 44–46 mm in diameter after 7 days of growth on MEA. The colony was white with flocculent hyphae in the center. The colony grew slowly on SNA (Figure 2A). The hyphae gradually tapered to apical and branched with a terminal sporangium. Sporangia were subglobose and smooth, and measured 1.4–4.6 × 1.3–3.3 μm in size. The chlamydospores were spherical and chain shaped, with diameters of 17–24 μm. Zygospores were not observed (Figure 2A).

L4: The colony diameter reached 61–63 mm after 7 days of cultivation on MEA. White cotton-like mycelia in the middle of the colony surface extended in all directions to form petal-like folds. After 7 days culture on PDA, the diameter of the strain reached 65–67 mm. The colony growth rate on SNA was slower than that on PDA and MEA. The diameter of the colony growing on SNA reached 28 mm after 7 days, and aerial mycelia were dispersed on the agar surface (Figure 2B). Sporogenesis was abundant with single-ended sporangium. Sporangia were spherical and 1.3–4.2 × 1.3–3.4 μm in size. Sporangia were globose to subglobose. No zygospores were seen (Figure 2B).

L5: The colony on PDA grew moderately fast, and the diameter reached 60–64 mm after 7 days of incubation at 30 °C. The front and back of the colony were white. As time goes on, the front side of the colony formed a concentric pattern like a flower; the reverse side of the colony turned slightly yellow and was irregularly zonate. The colony growth rate on MEA was slightly lower than that on PDA. After 7 days of culture at 30 °C, the colony diameter reached 55–57 cm. The colony grew slowly on SNA, and the diameter was only 22–28 mm after 7 days. Colonies were lighter in color and formed concentric patterns like flowers (Figure 2C). Sporangia were spherical and measured 2.6–5.5 × 2.2–4.6 μm. No zygospores were found (Figure 2C).

L12: The strain grew very fast on PDA; after 7 days of culture at 30 °C, the colony diameter was 75–79 mm with a cotton-like center and white edge. The opposite side of the colony gradually became yellowish-white. After 7 days of culture at 30 °C on MEA, the colony diameter reached 55 mm and presented a relatively neat round shape. The center of the colony was similar to that grown on PDA, with a distinct bulge. After 7 days of culture on SNA, the diameter of the strain was only 19–20 mm, with obvious aerial mycelia (Figure 2D). Sporangia were nearly globose, 2.5–5.4 × 2.0–5.0 μm in size. No zygospores were observed (Figure 2D).

Overall, the colonies of these isolates grew rapidly, especially on PDA, forming concentric patterns with flower-shaped radial growth, which are characteristic of the genus *Mortierella*. The growth temperatures of fungal isolates L3, L5, and L12 were determined to be between 4 and 30 °C, whereas that of isolate L4 was between 4 and 37 °C.

### 3.3. Effects of Different Phosphorus Sources on Phosphate-Solubilizing Capacity of Fungal Isolate L4

Using tricalcium phosphate as the sole phosphorus source, the available phosphorus contents of L3, L4, L5, and L12 were 95.14, 118.13, 113.88, and 91.75 mg/L, respectively. Among them, strain L4 showed the highest phosphorus solubility and was selected for further study. In the following study, the effects of different culture conditions including different P, C, and N sources, as well as initial pH on phosphate-solubilizing capacity of isolate L4 were investigated.

Firstly, Ca_3_(PO_4_)_2_, AlPO_4_, or FePO_4_ were used as the sole phosphorus source in the PKO liquid medium, and the phosphate-solubilizing capacity of fungal isolate L4 was further determined. When Ca_3_(PO_4_)_2_ was used as the phosphorus source, the release of soluble phosphorus increased sharply on the third day (Figure 3A). Throughout the culture period, the pH of the medium showed a trend of decreasing first and then slightly rising (Figure 3B). When FePO_4_ was employed as the phosphorus source, the soluble phosphorus content of this strain reached a relatively high level on the first day, and showed an upward trend in the following 1–3 days, and fluctuated slightly from 3 to 7 days (Figure 3A). Likewise, the pH of the medium decreased rapidly after 1 day of culture and then stabilized during the whole culture process (Figure 3B). Interestingly, when AlPO_4_ was used as the sole phosphorus source, the available phosphorus content decreased significantly on the second day, and then continued to increase and tended to stabilize at the later stage of culture (Figure 3A). However, soluble phosphorus levels were relatively low throughout the culture period. The pH value of the corresponding medium gradually dropped and then eventually stabilized (Figure 3B). Overall, among the three phosphorus sources, FePO_4_ exhibited the best phosphorus-dissolving effect in the early stage of culture, while Ca_3_(PO_4_)_2_ released the most soluble phosphorus in the middle and late stages of culture (Figure 3A).

### 3.4. Effects of Different Carbon and Nitrogen Sources and Initial pH on Phosphate-Solubilizing Capacity of Fungal Isolate L4

To evaluate the effects of different carbon and nitrogen sources on the phosphorus-dissolving capacity of the fungal isolate, glucose and ammonium sulfate were substituted in the same proportion in PKO medium by different carbon and nitrogen sources, respectively. The results showed that strain L4 with glucose as the carbon source had the highest soluble phosphorus content, followed by starch, maltose, and sucrose (Figure 4A). When glucose was utilized as the carbon source, the pH of the culture medium was the lowest, which was up to 5.08, and the soluble phosphorus concentration in the fermentation broth reached 123.72 mg/L. When mannitol was used as the carbon source, no soluble phosphorus was detected in the culture medium, and the phosphorus-dissolving capacity was associated with the pH of the culture medium (Figure 4A). Similarly, experiments with different nitrogen sources showed that isolate L4 had phosphorus-solubilizing capacity under various nitrogen source culture conditions. When ammonium chloride, ammonium sulfate, and peptone were used as nitrogen sources, respectively, the concentration of soluble phosphorus was higher, suggesting that ammonia nitrogen was more suitable for releasing soluble phosphorus than nitrate nitrogen. In addition, the final pH in the liquid medium varied significantly with different nitrogen sources: when peptone was used as the nitrogen source, the pH was the lowest, which could reach 5.46; the pH in the medium with ammonium nitrogen as a nitrogen source was lower than that with nitrate nitrogen (Figure 4B).

The capacity to dissolve calcium phosphate at different initial pH was further compared. When the pH was 9, the phosphorus-dissolving capacity was inhibited, the soluble phosphorus content was only 29.78 mg/L, and the final pH dropped substantially. When the initial pH was 5, there was no significant difference in the pH of the strain at the later stage of culture, and the relatively highest phosphorus-dissolving capacity was obtained (Figure 4C).

### 3.5. Analysis of Organic Acid and Phosphatase Activity of Fungal Isolate L4

Phosphorus-solubilizing microorganisms mainly rely on organic acid secretion [35]. However, not all organic acids generated by microorganisms can dissolve mineralized phosphate, and the structure and type of organic acids influence their capacity to dissolve phosphorus. The results of HPLC showed that the composition and content of organic acids in the culture supernatant secreted by strain L4 varied with the culture time (Figure 5). After 1 day of culture, four distinct organic acids were detected in the supernatant of PKO medium, including oxalic acid, lactic acid, acetic acid, and succinic acid, among which succinic acid was the absolute dominant, and its content reached 83.77 mg/L. Three additional organic acids appeared in liquid medium after 3 days of culture: tartaric acid, malic acid, and citric acid. Although succinic acid was still the most abundant, its concentration decreased significantly. After 5 days of cultivation, lactic acid (78.34 mg/L) secreted by L4 gradually increased and became the dominant organic acid in the culture supernatant, whereas succinic acid concentration continued to decline. After 7 days of culture, the L4 strain produced the most lactic acid, followed by acetic acid, succinic acid, and malic acid, with the first three showing a decreasing trend. In addition, although oxalic acid and tartaric acid contents gradually increased, they remained low throughout the culture period (Figure 5). These results suggested that a mixture of various organic acids has a faster and more efficient phosphate-dissolving capacity. In this case, it seems that the mixed succinic acid, lactic acid, and acetic acid potentially play a major role in the dissolution of phosphorus.

Enzymatic hydrolysis is one of the main ways of organophosphorus conversion, and phosphatase can decompose or transform phosphorus-containing organic matters and accelerate the release of soluble phosphorus. In this study, using PNPP as a substrate, the extracellular acid phosphatase and alkaline phosphatase produced by isolate L4 in the culture process were determined to investigate the possibility of enzyme participation (Figure 6). The results showed that both alkaline phosphatase and acid phosphatase had the highest enzyme activity at the early stage of culture (24 h), and the activity of alkaline phosphatase (154.72 μmol/(L·h)) was higher than that of acid phosphatase (120.99 μmol/(L·h)). However, during the subsequent culture period, the activity of alkaline phosphatase firstly decreased sharply, and then gradually stabilized and eventually rose somewhat, while the activity of acid phosphatase decreased slightly and then gradually increased. Therefore, in the middle and late period of culture (72–120 h), the activity of acid phosphatase exceeded that of alkaline phosphatase.

## 4. Discussion

Phosphorus is an essential nutrient for plants. PSMs have the ability to convert insoluble phosphorus in soil and enhance the release of available phosphorus. In this study, four phosphorus-solubilizing fungi were isolated from the rhizosphere soil of a poplar plantation in China. Based on phylogenetic analysis, they were all identified as *Mortierella* spp., with typical *Mortierella* morphological characteristics, such as the formation of rosette pattern colonies.

The capacity of PSMs to transform insoluble organophosphates and inorganic phosphorus is related to soil nutrient richness and physiological growth status [36]. It is also affected by a variety of other parameters, including culture time, vitamin and micronutrient supplementation, and the initial concentration of soluble phosphorus [37]. Studies have shown that fungi-induced acidification of phosphate ores is determined by carbon and nitrogen sources and their concentrations [38]. It was found that in highly weathered soils with high phosphorus adsorption capacity, *Mortierella* strains alone did not boost phosphorus uptake, but when combined with AM mycorrhiza, they could partially overcome phosphorus remineralization [39]. PSMs dissolve inorganic phosphorus mostly by the production of organic acids [40,41] and the release of protons [42,43], which is the main phosphorus-dissolving mechanism. However, little is known about the mechanism of phosphorus solubilization of *Mortierella*.

Among the four isolates, the L4 strain was selected to further explore the influence of diverse nutritional circumstances and initial pH on phosphorus-dissolving capacity due to its relatively high capacity to release soluble phosphorus. Overall, the content of available phosphorus produced by L4 during culture rose initially and subsequently stabilized. In this experiment, the highest content of dissolved phosphorus changed in the sequence Ca_3_(PO_4_)_2_ > FePO_4_ > AlPO_4_, which might be related to the different solubility of phosphorus sources in the liquid medium. This result is inconsistent with previous studies that tested the solubility of several fungal strains on Ca_3_(PO_4_)_2_, AlPO_4_, FePO_4_, and phosphate rocks [44]. Other reports have found similar results, and they described this phenomenon, attributing it to the high solubility of calcium phosphate in the culture medium [45]. Interestingly, when FePO_4_ was exploited as a phosphorus source, however, a higher soluble phosphorus concentration was detected after 1 day of culture, which may be related to the rapid decline in the pH of the liquid medium to an extremely acidic environment. In addition, when AlPO_4_ was used as the phosphorus source, soluble phosphorus content decreased dramatically after 2–3 days of culture, possibly because the soluble phosphorus required by the L4 strain for its own growth surpassed the release of soluble phosphorus induced by this strain during this period.

Both carbon and nitrogen sources have been proven in studies to have a considerable influence on the phosphorus-solubilizing capacity of PSMs. According to 11 different carbon source detection experiments, *Aspergillus aculeatus* exhibited the maximum phosphorus-dissolving efficiency when arabinose was employed as a carbon source; however, when sucrose was utilized as a carbon source, the phosphorus-dissolving capacity was relatively low [46]. Another study also found that strain G8 had the strongest phosphorus-dissolving effect when using glucose as the carbon source, and the lowest soluble phosphorus content when using starch as the carbon source [47]. In this study, the available phosphorus content of the rhizosphere isolate L4 was the highest when the carbon source was glucose, while the soluble phosphorus was negligible when the carbon source was mannitol, indicating that mannitol was not conducive to the dissolution of phosphorus. The results of this study were not completely consistent with those of previous studies, which might be attributed to remarkable differences in the utilization of various carbon sources by different PSMs. In addition, our study also found that strain L4 had the highest available phosphorus content when ammonium sulfate was used as the nitrogen source, which was perfectly consistent with some previous studies, for example, the use of ammonium nitrogen was more likely to dissolve insoluble phosphorus than that of nitrate nitrogen [48].

As a critical factor in soil physical and chemical properties, pH plays an important role in the growth and function of soil microorganisms. In this study, when the pH was 9, the phosphorus-dissolving capacity was inhibited, and the final pH dropped substantially, indicating that isolate L4 may need to produce more acidic substances in order to obtain accessible phosphorus. When the initial pH was 5, there was no significant difference in pH for the strain at the later stage of culture, and the relatively highest phosphorus-dissolving capacity was obtained, indicating that the lower the initial pH in the medium, the higher the available phosphorus content released by the isolate (Figure 4C).

Microorganisms secrete a variety of organic acids during growth and metabolism. On the one hand, they have the ability to lower the pH of the environment and dissolve insoluble phosphorus compounds. On the other hand, organic acids can combine with Ca^2+^, Fe^3+^, Al^3+^, and other metal ions to release soluble phosphorus. It has been shown that the decrease in pH during phosphate dissolution is caused mostly by the production of organic acids [49]. Xiao et al. [50] also found that acidification appears to be the primary mechanism of phosphate dissolution in rocks, with three organic acids, including citric acid, oxalic acid, and gluconic acid, detected in the culture media inoculated with three phosphate-solubilizing isolates (*Penicillium expansum*, *Mucor ramosissimus*, and *Candida krissii*). In another study, oxalic acid and citric acid were found to be the two main organic acids secreted by the phosphorus-solubilizing fungus *Aspergillus niger* [51]. After 7 days of culture, oxalic acid had more substantial solubility of ferric phosphate and tricalcium phosphate than citric acid. In our study, L4 produced more succinic acid and lactic acid, suggesting that these two acids may play a leading role in the process of phosphorus dissolution. Our findings also implied that mixing diverse organic acids could potentially enhance the phosphorus dissolution effect. Recent research has further confirmed that phosphate can be solubilized by the production of organic acids [52].

Some studies have confirmed that the role of PSMs in phosphorus dissolution mechanism is also attributed to their ability to secrete extracellular phosphatase for organophosphorus conversion. UV mutagenesis showed that the phosphorus-dissolving capacity was related to the decrease in pH and the activities of acid phosphatase and phytase in the culture supernatant [53]. Phosphatases can be divided into acid phosphatases and alkaline phosphatases according to their optimal pH, both of which can be produced by PSMs in response to external circumstances [54]. In acidic soils, acid phosphatase is dominant, while alkaline phosphatase is more abundant in neutral and alkaline soils. In this study, phosphatase activity analysis revealed that both alkaline and acid phosphatase had the highest enzyme activity at the early stage of culture (24 h; pH > 7), and the activity of alkaline phosphatase was higher than that of acid phosphatase. After 3 days of culture with pH less than 6, acid phosphatase was found to be dominant, but the activity of both enzymes declined under acidic conditions. Interestingly, since the soil pH of the sampling site is alkaline (pH ≈ 8), alkaline phosphatase should be more predominant in soil organophosphorus conversion and soluble phosphorus release. Therefore, the high phosphatase activity of L4 isolated in this study under alkaline conditions has significant ecological implications for the poplar plantation ecosystem in which L4 is isolated.

Owing to the pollution caused by the overuse of phosphate fertilizer, biofertilizer has been an area of intense investigation. The microbial phosphorus solution approach is efficient, low-cost, and environmentally friendly. This study demonstrated that strain L4 of the genus Mortierella could dissolve various phosphorus sources and survive at temperatures ranging from 4 to 37 °C. In addition, strain L4 was able to produce oxalic acid on the very first day, and the concentration of oxalic acid steadily increased, hence potentially enhancing the phosphorus solubilization effect. Some studies have found that oxalic acid has a high potential solubilization effect on rock phosphate [55]. Oxalic acid is more effective than sulfuric acid and can release more phosphorus [56]. Therefore, *Mortierella* strains are anticipated to be utilized as a biofertilizer to promote plant growth. Nevertheless, at the same time, it is essential to consider all the potential possibilities and risks when using fungal biofertilizers [57].

## 5. Conclusions

In this study, the phosphorus-solubilizing capacity, phosphorus-solubilizing characteristics, and phosphorus-solubilizing mechanism of the selected phosphorus- dissolving fungi strains were investigated, in order to improve the utilization rate of phosphorus in the soil, overcoming a series of environmental problems caused by the massive application of phosphorus fertilizer, and providing a theoretical foundation for agricultural production and manufacturing of sustainable and ecological microbial fertilizers. Four phosphorus-dissolving fungi were isolated from rhizosphere soil of poplar plantations and identified as *Mortierella* spp. Further study of strain L4 with the highest phosphorus-solubilization capability revealed that its solubility of tricalcium phosphate was superior to that of ferric phosphate and aluminum phosphate. When glucose and ammonium sulfate were used as carbon and nitrogen sources, respectively, or when the initial pH was 5, the culture supernatant contained the most soluble phosphorus. The mechanism strain L4 employs to solubilize inorganic phosphorus is believed to depend primarily on the production of several organic acids, such as succinic acid, lactic acid, and acetic acid, which reduce the pH of the medium solution. Furthermore, the detected acid/alkaline phosphatase activity indicates that the phosphatase produced by this strain also plays a role in the conversion of organophosphorus and the release of soluble phosphorus.

## Figures and Tables

**Figure 1 microorganisms-10-02361-f001:**
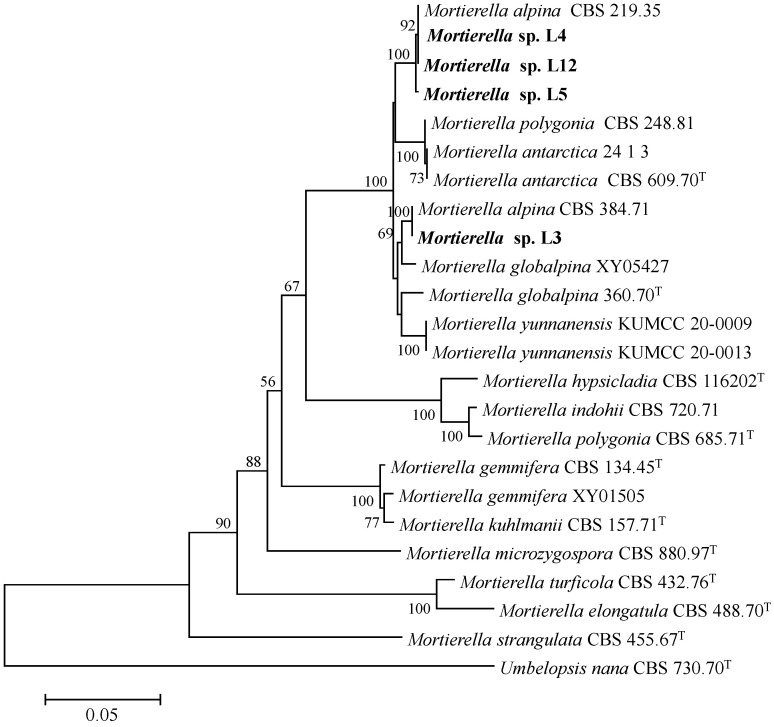
Neighbor-joining phylogenetic analysis based on the combined sequences of internal transcribed spacer (ITS) and large subunit (LSU) regions of the four fungal isolates. The phylogenetic tree was constructed using the MEGA 6 program. The supported values from the 1000 bootstrap copies are illustrated under their respective branches. The sequences obtained in this study are marked in bold.

**Figure 2 microorganisms-10-02361-f002:**
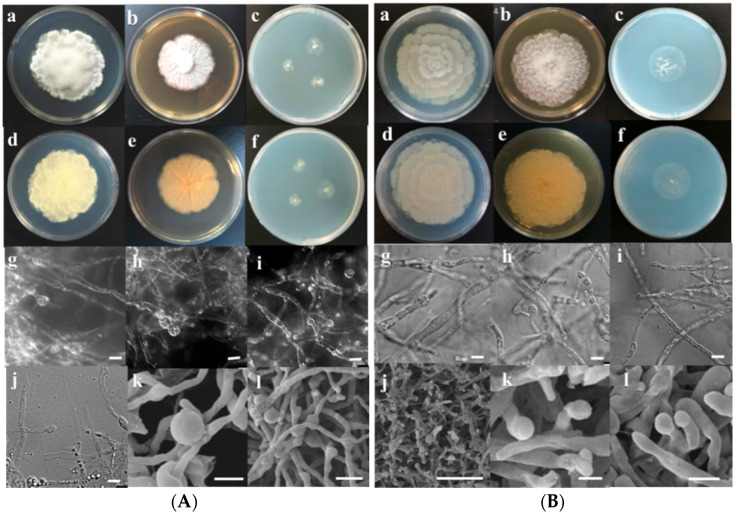
Morphology of the fungal isolates. (**A**) L3. Colonies grown on different media after 7 days at 30 °C, a–c: left to right, obverse PDA, MEA, and SNA. d–f: left to right, reverse PDA, MEA, and SNA. g–l: Microscopic and scanning electron microscopy of sporangia (scale bars: g–j, 20 μm; k, 5 μm; l, 10 μm). (**B**) L4. Colonies grown on different media after 7 days at 30 °C, a–c: left to right, obverse PDA, MEA, and SNA. d–f: left to right, reverse PDA, MEA, and SNA. g–l: Microscopic and scanning electron microscopy observation of sporangia (scale bars: g–i, 20 μm; j, 30 μm; k–l, 5 μm). (**C**) L5. Colonies grown on different media after 7 days at 30 °C, a–c: left to right, obverse PDA, MEA, and SNA. d–f: left to right, reverse PDA, MEA, and SNA. g–l: Microscopic and scanning electron microscopy of sporangia (scale bars: g–i, 20 μm; j, 20 μm; k–l, 5 μm). (**D**) L12. Colonies grown on different media after 7 days at 30 °C, a–c: left to right, obverse PDA, MEA, and SNA. d–f: left to right, reverse PDA, MEA, and SNA. g–l: Microscopic and scanning electron microscopy of sporangia (scale bars: g–i, 20 μm; j, 10 μm; k–l, 5 μm).

**Figure 3 microorganisms-10-02361-f003:**
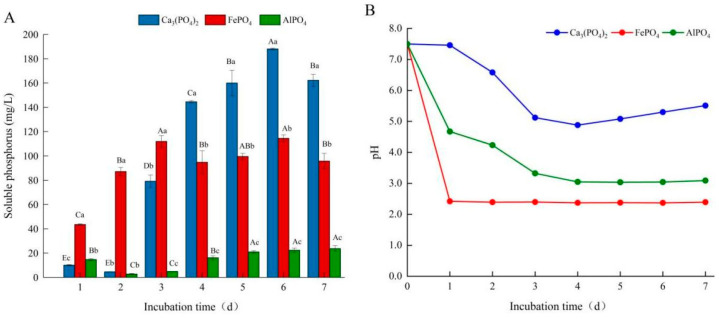
Changes in soluble phosphorus concentration (**A**) and pH value (**B**) in fermentation broth with different phosphorus sources. Different capital letters represent the significant (*p* < 0.05) difference of the soluble phosphorus concentration under the same phosphorus source at different cultivation times; different lowercase letters indicate that the soluble phosphorus content in the culture supernatant of different phosphorus sources is significantly (*p* < 0.05) different under the same culture time.

**Figure 4 microorganisms-10-02361-f004:**
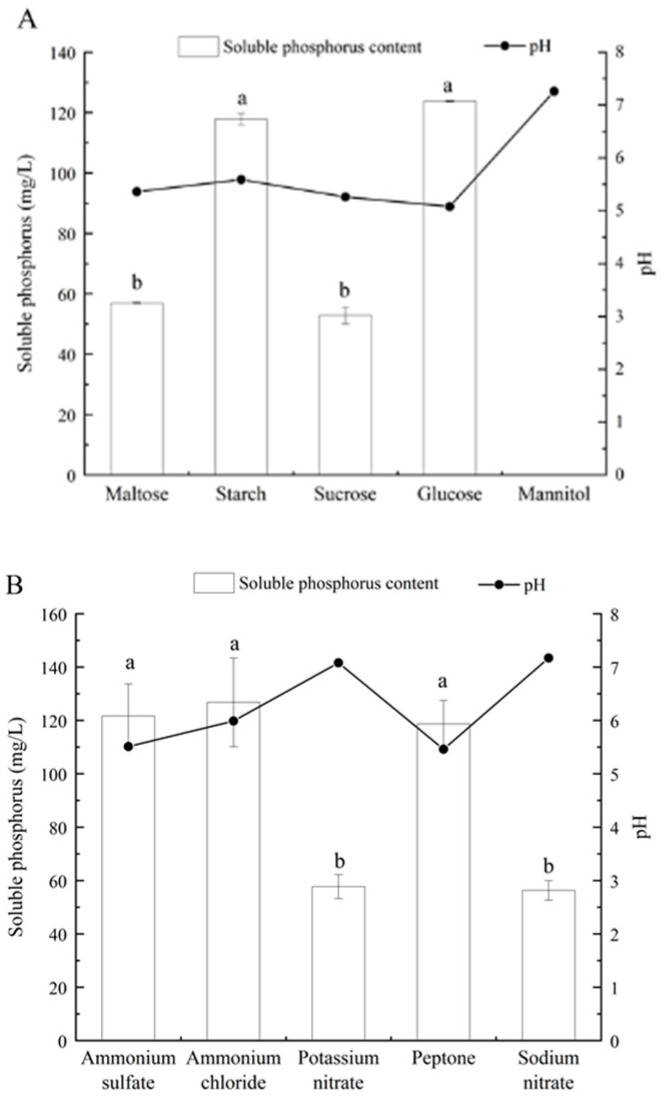
Effect of different carbon (**A**) and nitrogen sources (**B**), and initial pH (**C**) on the phosphate-solubilizing capacity of fungal isolate L4. Different lowercase letters indicate significant (*p* < 0.05) difference between treatments.

**Figure 5 microorganisms-10-02361-f005:**
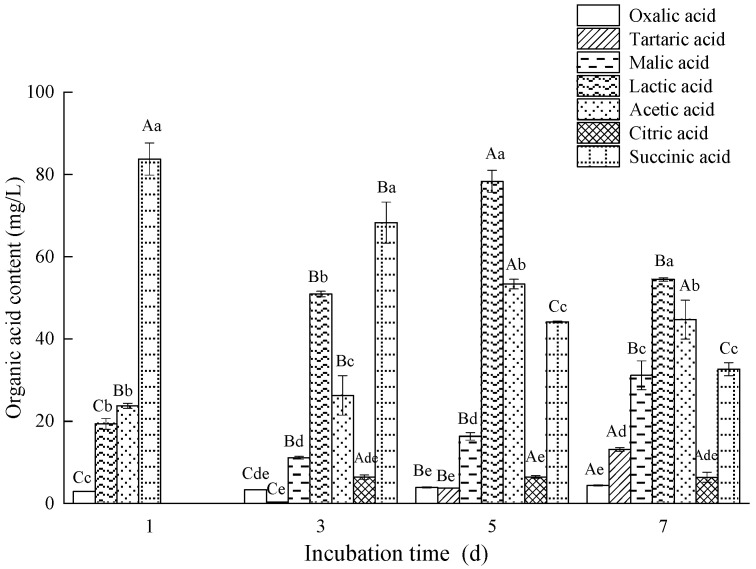
Concentration of various organic acids secreted by L4 at different culture times. Different capital letters indicate a significant (*p* < 0.05) difference in the content of the same organic acid; different lowercase letters indicate significant (*p* < 0.05) differences in different organic acid contents in the same day.

**Figure 6 microorganisms-10-02361-f006:**
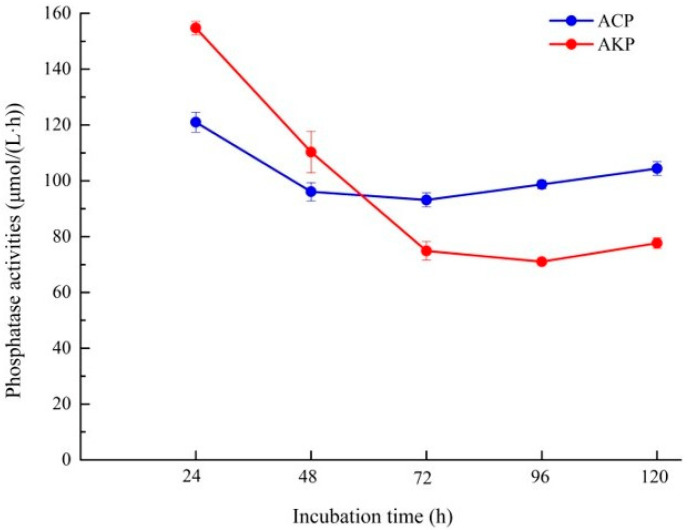
Phosphatase activity of fungal isolate L4. ACP, acid phosphatase; AKP, alkaline phosphatase.

**Table 1 microorganisms-10-02361-t001:** Strains used in the molecular phylogenetic analysis in this study.

Species	Collection Isolate	GenBank Accession Number	
	Number	28S	ITS
L3	CGMCC 3.26004	ON045511	ON038715
L4	CGMCC 3.26005	ON045513	ON038743
L5	CGMCC 3.26006	ON045526	ON038747
L12	CGMCC 3.26007	ON045523	ON038748
*M. antarctica*	CBS 609.70^T^	NG_042563.1	NR_111580.1
*M. antarctica*	24_1_3	MT521860.1	MT521806.1
*M. yunnanensis*	KUMCC20-0009	NG_075333.1	NR_172421.1
*M. yunnanensis*	KUMCC20-0013	MT032143.1	MT031918.1
*M. hypsicladia*	CBS 116202^T^	NG_042547.1	NR_111563.1
*M. gemmifera*	CBS 134.45^T^	NG_042543.1	NR_111559.1
*M. gemmifera*	XY01505	MT521851.1	MT521798.1
*M. polygonia*	CBS 685.71^T^	NG_042546.1	NR_111562.1
*M. polygonia*	CBS 248.81	JX976145.1	JX975891.1
*M. globalpina*	XY05427	MT521812.1	MT521758.1
*M. globalpina*	CBS 360.70^T^	NG_064079.1	NR_160121.1
*M. microzygospora*	CBS 880.97^T^	NG_042553.1	NR_111569.1
*M. indohii*	CBS 720.71	NG_042545.1	NR_111561.1
*M. kuhlmanii*	CBS 157.71^T^	NG_042544.1	NR_111560.1
*M. alpina*	CBS 219.35	JX976018.1	MH867163.1
*M. alpina*	CBS 384.71C	MH860175.1	JX976154.1
*M. turficola*	CBS 432.76^T^	NG_042566.1	NR_111583.1
*M. elongatula*	CBS 488.70^T^	NG_042565.1	NR_111582.1
*M. strangulata*	CBS 455.67^T^	NG_057902.1	JX975997.1
*Umbelopsis nana*	CBS 730.70^T^	NG_058036.1	MH859921.1

## Data Availability

Not applicable.

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
