# Peer review of "Phosphorus-Solubilizing Capacity of Mortierella Species Isolated from Rhizosphere Soil of a Poplar Plantation"

_microorganisms, 2022, doi:10.3390/microorganisms10122361_

Round 1

Reviewer 1 Report

It is interesting for the manuscript to explore the role of phosphorus-solubilizing fungi in converting insoluble phosphorus into available phosphorus in soil, but the Abstract should clearly state the results in your manuscript.

I feel confuse about the meaning of paragraph 3 in Introduction section and I think it is not appropriate in here. In the Introduction a number of references are a bit old, there are many recent references on PSMs.

How the L3, L4, L5, and L12 isolated from the soil, please specify in the materials and methods.

The Results section in the present manuscript is large parts unprecise, such as “relatively high level”, “then eventually stabilized” and so on. Also, the statements of experimental methods and discussion of the results are included in Results section. Further, please put Figure 1 in front of Table 1, which will make it easier for the readers to understand. In Figure 1, the statement of “The sequences obtained in this study are marked in bold red.” Bold red? Please check.

The discussion seemed should be deeply revised. The statement of “PSMs dissolve inorganic phosphorus mostly by the production of organic acids and the release of protons, which is the main phosphorous dissolving mechanisms.”, it seemed the mechanism of phosphorous release by PSMs already have explored. And What the main findings of this study? Besides, were some discussions the speculation of authors or from the literature or from this study? If from literatures or this study, please note the references or figures/tables (e. g. “In this experiment, the highest content of dissolved phosphorus changed in the sequence Ca3(PO4)2>FePO4>AlPO4, which might be related to the different solubility of phosphorus sources in the liquid medium, or it could be due to the different quantities and types of organic acids and other substances secreted by microorganisms.”).

It should be addressed the purpose of this study in the Conclusion.

Author Response

Reviewer 1

It is interesting for the manuscript to explore the role of phosphorus-solubilizing fungi in converting insoluble phosphorus into available phosphorus in soil, but the Abstract should clearly state the results in your manuscript.

Authors’ response: Thanks for your detail comments, some specific results about phosphorus-solubilizing activity was given following your recommendation (L23-29).

I feel confuse about the meaning of paragraph 3 in Introduction section and I think it is not appropriate in here. In the Introduction a number of references are a bit old, there are many recent references on PSMs.

Authors’ response: 

- This paragraph is mostly focused on the taxonomic area, and since this study includes a portion of the taxonomic component, we believe that it needs some discussion of the genus Mortierella in order to help readers quickly understand the substance of our research.

- We are very grateful for the detailed comments you provided, and we have made the changes you suggested to the references that were listed in the part of the introduction that they were in.

How the L3, L4, L5, and L12 isolated from the soil, please specify in the materials and methods.

Authors’ response: More specific isolation methods were given in the text according to your comments (L89-92).

The Results section in the present manuscript is large parts unprecise, such as “relatively high level”, “then eventually stabilized” and so on. Also, the statements of experimental methods and discussion of the results are included in Results section. Further, please put Figure 1 in front of Table 1, which will make it easier for the readers to understand. In Figure 1, the statement of “The sequences obtained in this study are marked in bold red.” Bold red? Please check.

Authors’ response:

- We thanks a lot the reviewer’s advices, and we tried to avoid inaccurate wording like “relatively high level” and “then eventually stabilized”, but no specific scientific evidence could be discovered to replace them.

- Some repeated statements were modified following your advices (L282-284; L297-299; L391-392; L400-407).

- Figure 1 was placed in front of Table 1 following your suggestion.

- We apologize for the confusion, we corrected it to “in bold” (Figure 1).

The discussion seemed should be deeply revised. The statement of “PSMs dissolve inorganic phosphorus mostly by the production of organic acids and the release of protons, which is the main phosphorous dissolving mechanisms.”, it seemed the mechanism of phosphorous release by PSMs already have explored. And What the main findings of this study? Besides, were some discussions the speculation of authors or from the literature or from this study? If from literatures or this study, please note the references or figures/tables (e. g. “In this experiment, the highest content of dissolved phosphorus changed in the sequence Ca3(PO4)2>FePO4>AlPO4, which might be related to the different solubility of phosphorus sources in the liquid medium, or it could be due to the different quantities and types of organic acids and other substances secreted by microorganisms.”).

Authors’ response: There are few studies on the mechanism of phosphorus solubilization of Mortierella. This study mainly explores the organic acids produced by Mortierella as a phosphorus-solubilizing fungus to dissolve phosphorus and the possible role of enzymatic hydrolysis in phosphorus solubilization. Based on your recommendations, the discussion part was revised (L363-364; L391-392; L 400-407; L442-453).

It should be addressed the purpose of this study in the Conclusion.

Authors’ response: Thanks for your recommendation, and purpose of the research has been given in the part of conclusion (L455-460).

We would like to express our gratitude for the insightful suggestions that you have provided regarding our research paper. These remarks will be included into an effort to enhance the overall quality of the work. We tried our best to improve the manuscript and made all changes that reviewers mentioned in the manuscript (marked blue). These changes will not influence the comment and framework of the paper. We appreciate for your detail and kind work earnestly, and hope that the corrections made here will meet with approval. Once again, thank you very much for your comments and suggestions.

Yours sincerely

Prof. Feng-Jie Jin

Nanjing Forestry University

Reviewer 2 Report

Dear Authors,

Thank you for the opportunity to review your manuscript "Phosphorus solubilizing capacity of Mortierella species isolated from rhizosphere soil of a poplar plantation." It is a novel study with definite relevance to agriculture and soil science. The results are sound and open a whole new venue for further research in this direction. However, your manuscript cannot be presented to the broader audience in its current form. Your isolates belong to different genera of Mortierellomycotina, therefore the whole phylogeny and descriptions of your isolates and their properties are in acute need of thorough revision. Also, Mortierella isabellina is now in the genus Umbellopsis and doesn't belong to the fungal group you study. Please check all species you mention in Index Fungorum and change their scientific names accordingly. Also, you might find useful the paper "Resolving the Mortierellaceae phylogeny through synthesis of multi-gene phylogenetics and phylogenomics" dealing with recent changes in this fungal group.

I have another concern regarding the figures. Is any other way to rephrase the statements like "different lowercase letters indicate significant (P < 0.05) difference between different organic acids at the same culture time?" It is difficult to comprehend such repeatable statements, perhaps I am not familiar with such a presentation of statistical analysis. Also, figure 6 contains some information I was unable to locate, namely: "ACP, acid phosphatase; AKP, alkaline phosphatase." The title of the x-axes "Incubation time(d)" definitely needs space before () because it reads differently otherwise.

Despite these flaws, your manuscript definitely has significant merit and can be improved. I suggest a major revision.

Respectfully,

Reviewer.

Author Response

Reviewer 2

Thank you for the opportunity to review your manuscript "Phosphorus solubilizing capacity of Mortierella species isolated from rhizosphere soil of a poplar plantation." It is a novel study with definite relevance to agriculture and soil science. The results are sound and open a whole new venue for further research in this direction. However, your manuscript cannot be presented to the broader audience in its current form. Your isolates belong to different genera of Mortierellomycotina, therefore the whole phylogeny and descriptions of your isolates and their properties are in acute need of thorough revision. Also, Mortierella isabellina is now in the genus Umbellopsis and doesn't belong to the fungal group you study. Please check all species you mention in Index Fungorum and change their scientific names accordingly. Also, you might find useful the paper "Resolving the Mortierellaceae phylogeny through synthesis of multi-gene phylogenetics and phylogenomics" dealing with recent changes in this fungal group.

Authors’ response:

- We are very grateful for the information that you have supplied. In addition, we are sorry that the reclassification of Mortierella isabellina was overlooked. The species Mortierella isabellina has been removed from the genus Mortierella, and the number of species that belong to the genus Mortierella has been adjusted in response to the feedback that you provided (L58-62).

- The phylogenetic tree of combined sequences has been reconstructed, and more species from the genus Mortierella have been added to it.

I have another concern regarding the figures. Is any other way to rephrase the statements like "different lowercase letters indicate significant (P < 0.05) difference between different organic acids at the same culture time?" It is difficult to comprehend such repeatable statements, perhaps I am not familiar with such a presentation of statistical analysis. Also, figure 6 contains some information I was unable to locate, namely: "ACP, acid phosphatase; AKP, alkaline phosphatase." The title of the x-axes "Incubation time(d)" definitely needs space before () because it reads differently otherwise.

Authors’ response:  The legend of figure 5 has been updated as following your suggestions, for which the authors are very grateful. We apologize for the duplicate figures in figures 5 and 6. We believe that it occurred during template formatting. Figure 6 was replaced with a new version.

Despite these flaws, your manuscript definitely has significant merit and can be improved. I suggest a major revision.

We appreciate your positive feedback on the submitted article. The manuscript was revised in accordance with your recommendations. These modifications will not affect the paper's discussion or structure. We sincerely appreciate your attention to detail and kindness, and we hope that the corrections made here will be accepted. Thank you again for your thoughtful comments and suggestions.

Yours sincerely

Prof. Feng-Jie Jin

Nanjing Forestry University

Round 2

Reviewer 2 Report

Dear Authors,

Thank you for another opportunity to review your manuscript. You made considerable improvements. I don't have any concerns regarding the material you present except one. In Figure 1, the species Mortierella epicladia, M. horticola, and M. humilis belong to the genus Podila. Since you correctly identified the placement of your specimens in the genus Mortierella and your paper doesn't deal with the taxonomy of Mortierellas and related genera, it is not really a big deal. I would just prune this branch from the tree to avoid unnecessary comments or suggestions. Except for this minor remark, your manuscript is ready to be published, I believe.

Respectfully,

Reviewer.

Author Response

Reviewer 2

Thank you for another opportunity to review your manuscript. You made considerable improvements. I don't have any concerns regarding the material you present except one. In Figure 1, the species Mortierella epicladia, M. horticola, and M. humilis belong to the genus Podila. Since you correctly identified the placement of your specimens in the genus Mortierella and your paper doesn't deal with the taxonomy of Mortierella and related genera, it is not really a big deal. I would just prune this branch from the tree to avoid unnecessary comments or suggestions. Except for this minor remark, your manuscript is ready to be published, I believe.

Respectfully,

Reviewer.

The authors' response: We really appreciate your remarks about the taxonomy of Mortierella, and we apologize once more for any misunderstandings or shortcomings in the field of taxonomy that may have resulted from our work. Figure 1 was replaced with updated version, and the information regarding the three sequences you mentioned above has been removed from Table 1.  

Yours sincerely

Prof. Feng-Jie Jin

Nanjing Forestry University